# Action-Conditioned Transformers for Decentralized Multi-Agent World Models

## Abstract

Multi-agent reinforcement learning (MARL) has achieved strong results on large-scale decision making, yet most methods are model-free, limiting sample efficiency and making coordination harder as teammates' policies evolve during training. Model-based reinforcement learning can reduce data usage, but planning and search scale poorly with joint action spaces. We introduce MACT, a decentralized action-conditioned Transformer world model for long-horizon cooperative control. Each agent processes discretized observation-action tokens with a shared Transformer, while a single cross-agent Perceiver step provides global context under centralized training and decentralized execution. MACT targets long-horizon coordination by coupling Perceiver-derived global context with an action-conditioned contrastive objective that predicts future latent representations over a short horizon conditioned on planned actions. On the StarCraft Multi-Agent Challenge, MACT achieves the strongest aggregate performance among 6 model-free and model-based baselines under matched low-data training budgets. Across 12 maps, including 2 SuperHard scenarios, MACT attains the best mean and median win rates, and ablations show that multi-step prediction and per-agent action conditioning are central to its gains.

## 1 Introduction

Model-free multi-agent algorithms such as QMIX (Rashid et al., 2020), QPLEX (Wang et al., 2021), and MAPPO (Yu et al., 2022) can achieve robust long-term returns, but they often do so at the cost of millions of environment interactions. Two structural factors drive this sample hunger: the joint observation-action space grows rapidly as team size increases (Liu et al., 2024), and non-stationarity emerges because each agent's data distribution changes in response to its teammates' evolving policies (Gronauer & Diepold, 2022). For example, consider a focus-fire movement in the StarCraft Multi-Agent Challenge (SMAC) (Samvelyan et al., 2019), where a group of units must be commanded to attack a single enemy to eliminate it faster. Success depends on understanding delayed consequences of the team's joint actions. This is the type of long-horizon reasoning that models trained only on one-step prediction can have difficulty capturing.

In single-agent settings, model-based reinforcement learning addresses related sample-efficiency issues by training a latent world model (Ha & Schmidhuber, 2018) that can be rolled forward in imagination, replacing expensive real transitions with synthetic ones, as in Dreamer (Hafner et al., 2020). The Transformer-based successor TWISTER (Burchi & Timofte, 2025), together with the domain-robust DreamerV3 (Hafner et al., 2025), shows that accurate latent dynamics can greatly reduce sample cost when the objective encourages multi-step predictive structure. Transferring this promise to multi-agent scenarios is difficult because explicit planning over the joint action space scales poorly. Existing multi-agent world models therefore tend to learn a latent model and train policies through imagined rollouts rather than performing online search. MAMBA (Egorov & Shpilman, 2022) uses a shared recurrent latent state, while MARIE (Zhang et al., 2025) improves scalability by assigning each agent local token dynamics and injecting joint context through a lightweight Perceiver cross-attention layer. However, these approaches remain primarily supervised by one-step reconstruction losses, which can bias representations toward short-term correlations and make long-horizon coordination more fragile.

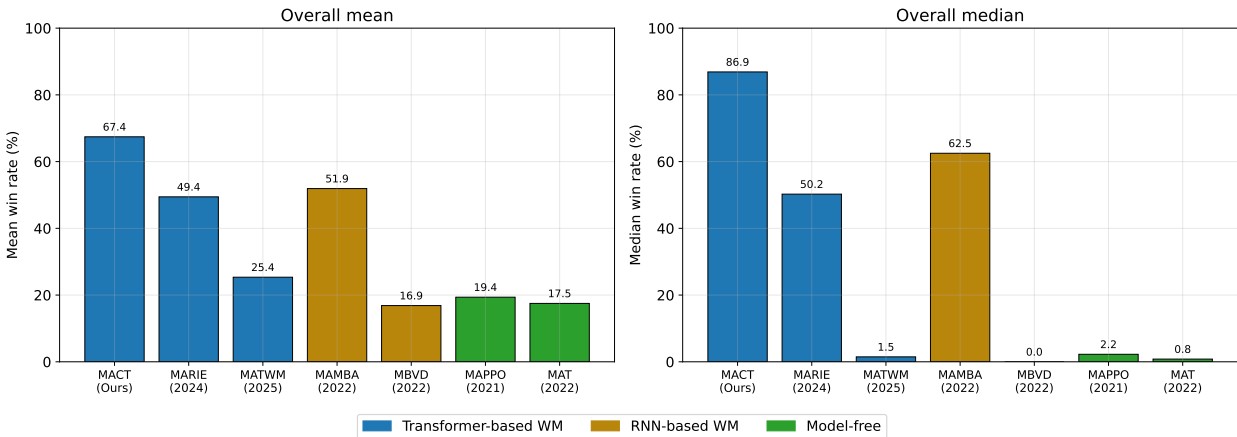

Figure 1: Mean (left) and median (right) win rates evaluation across all SMAC environments. Bars are color-coded by learning methodology.

TWISTER showed in the single-agent domain that a Transformer equipped with action-conditioned contrastive predictive coding (AC-CPC) can exploit its representational capacity more fully than a purely one-step objective. Instead of only reconstructing the next latent state, the model predicts a sequence of future latents conditioned on an action window from the sampled replay segment and learns by contrasting the true future against negatives in the batch. This long-horizon objective encourages temporally predictive abstractions over multiple steps. Bringing this idea to cooperative MARL is not plug-and-play: one must decide *what* representation should be predicted under centralized training and decentralized execution (CTDE), and *which* actions should condition that prediction, while avoiding trivial token matching and preserving scalable cross-agent context.

Our goal is to bring the benefits of model-based learning to cooperative control under tight data budgets without sacrificing simplicity or scale. We present **MACT**, a **M**ulti-agent **A**ction-**C**onditioned **T**ransformer that builds on the tokenized Transformer/Perceiver world-model backbone of MARIE (Zhang et al., 2025) and changes its training signal. MACT predicts future team-conditioned Perceiver latents from each agent's local latent and a short per-agent action window, using an augmented future view as the positive target. This ties an agent's planned actions to its future team-conditioned representation while avoiding exact token matching.

To enable a controlled and meaningful assessment of model-based and model-free multi-agent reinforcement learning methods, we evaluate MACT, MARIE, MATWM, MAMBA, MBVD, MAPPO, and MAT under a unified experimental protocol, using identical per-map interaction budgets and consistent evaluation criteria. Figure 1 summarizes the resulting aggregate behavior: under this unified protocol, MACT achieves the best mean and median performance across the 12-map benchmark. The closest aggregate baseline is MAMBA rather than MATWM, and the SuperHard maps further clarify both strengths and limitations: MACT performs well on `corridor`, whereas `6h_vs_8z` remains unsolved by all methods at this budget.

Our contributions are described as follow:

- We design an action-conditioned, multi-step contrastive learning objective for decentralized MARL by resolving CTDE-specific action-conditioning and target choices: each agent predicts future team-conditioned Perceiver latents from its local latent and a short per-agent action window, using an augmented future view as positives to avoid trivial matching.

- We provide a unified same-protocol SMAC evaluation of MACT and six baselines, with per-seed uncertainty, rliable-style aggregate metrics, and overlaid learning curves through 12 SMAC environments. While diagnostic analyses showing that multi-step horizons and per-agent action conditioning are central while augmentation remains map-dependent.

## 2 Related Work

Model-free MARL has progressed through value-factorization methods (VDN (Sunehag et al., 2018), QMIX (Rashid et al., 2020), QPLEX (Wang et al., 2021)), policy-gradient variants (MAPPO (Yu et al., 2022), HAPPO (Kuba et al., 2022)), and sequence-model policies such as Multi-Agent Transformer (MAT) (Wen et al., 2022). All of them follow the CTDE recipe in different forms: global information can be used during learning, but each agent runs a local policy at test time. Because these methods do not learn an environment model, every useful joint configuration must still be sampled through real interaction. This is the obstacle that our world-model approach seeks to overcome.

Single-agent model-based RL pre-trains a generative model of environment dynamics and then improves a policy inside that model. Early versions such as SimPLe (Kaiser et al., 2019) employed learned video prediction, while Dreamer (Hafner et al., 2020) switched to a recurrent state-space model and latent imagination. DreamerV3 (Hafner et al., 2025) refined the recipe, achieving domain robustness without per-task tuning. Several groups replaced RNNs with Transformers to exploit parallel training: IRIS (Micheli et al., 2023) maps each frame to a grid of VQ-VAE (van den Oord et al., 2017) tokens and processes the result with a spatial-temporal Transformer. TWM (Robine et al., 2023) concatenates observation, action, and reward tokens and trains a Transformer-XL. STORM (Zhang et al., 2023) adds stochastic latent variables to a GPT-like backbone and reports strong Atari-100k results. These works validate Transformers as world-model cores but still commonly rely on next-step prediction, which can underuse long-horizon structure.

Contrastive objectives address this limitation. CPC (van den Oord et al., 2018) maximizes mutual information between present and future representations by contrasting the true future against negatives. In visual RL, CURL (Laskin et al., 2020) treats different data-augmented views of the same observation as a positive pair to learn spatial features, but it does not directly model the temporal and action-conditioned nature of control. Building on temporal and action-driven features, TACO (Zheng et al., 2023) predicts future states conditioned on future action sequences, learning both state and action representations. TWISTER (Burchi & Timofte, 2025) pairs AC-CPC with a Transformer world model, showing that long-horizon objectives can unlock Transformer capacity in low-data regimes. MACT adapts this action-conditioned, multi-step principle to cooperative CTDE, where future representations must account for both an agent's own actions and the team context.

Multi-agent world models face an additional scalability challenge: the joint observation-action space grows exponentially with team size. MAMBA (Egorov & Shpilman, 2022) adapts Dreamer to SMAC but keeps a single shared latent state, which limits scalability. MARIE (Zhang et al., 2025) distributes token dynamics over agent-specific histories and injects global context through one step of Perceiver cross-attention. MATWM (Deihim et al., 2025) augments a Transformer world model with a teammate predictor module, while MBVD (Xu et al., 2022) combines learned models with value decomposition. MACT belongs to this world-model family, but its central change is the training signal: instead of relying only on one-step reconstruction, it adds a per-agent action-conditioned contrastive objective that explicitly trains multi-step team-conditioned representations.

## 3 Methodology

In this section we describe the MACT world-model architecture, the training objectives applied to that architecture, and the imagination-based actor–critic procedure that consumes the learned world model. To separate attribution from the new contribution, we explicitly use MARIE's tokenized per-agent Transformer/Perceiver backbone (Zhang et al., 2025): the vector tokenizer, local Transformer dynamics, Perceiver aggregation, one-step prediction heads, and imagination-based actor–critic follow that pipeline. We restate these components to make the notation self-contained. The new part of MACT is the action-conditioned multi-step contrastive objective in Eq. equation 6 and Eq. equation 8, including the CTDE-specific target and action-conditioning choices.

On SMAC environments (Samvelyan et al., 2019), each map is modeled as a Dec-POMDP $\langle \mathcal{S}, \mathcal{A}^{1:N}, P, R, \Omega^{1:N}, \gamma \rangle$. At time $t$, every allied unit $i \in \{1{:}N\}$ receives a structured vector observation $o_t^i \in \mathbb{R}^{d_o}$ of modest dimensionality, around $d_o \approx 70$ for map `3s_vs_5z`, and chooses a discrete action $a_t^i \in \mathcal{A}^i$

Table 1: Comparison of MACT with leading multi-agent world-model architectures.

| Aspect | MACT (Ours) | MARIE | MATWM | MAMBA | MBVD |
|---|---|---|---|---|---|
| Backbone Architecture | Transformer | Transformer | Transformer | GRU/RSSM | GRU/VAE |
| Latent Representation | VQ-VAE | VQ-VAE | Categorical VAE | Categorical VAE | VAE |
| Critic Type | Centralized | Centralized | Semi-centralized | Centralized | Centralized |
| World Model Type | Hybrid | Hybrid | Decentralized | Centralized | Centralized |
| Agent Training Strategy | PPO-style | PPO-style | DreamerV3-style | PPO-style | Deep Q-learning |
| Prediction Horizon | $K_{\mathrm{cpc}}$=8 steps | Next state | Next state | Next state | Next state |

from a finite categorical set such as movement, attack, or stop. Although each local input is compact, the joint observation space $\Omega^{1:N}=\Omega^1\times\ldots\times\Omega^N$ still grows exponentially with $N$. On SMAC we assume a shared discrete action set across agents, i.e., $\mathcal{A}^i = \mathcal{A}$ for all $i \in \{1{:}N\}$, and thus $|\mathcal{A}^i| = |\mathcal{A}|$. We index actions as integers $a_t^i \in \{1{:}|\mathcal{A}|\}$, and use the corresponding one-hot vectors $\mathrm{onehot}(a_t^i) \in \{0,1\}^{|\mathcal{A}|}$ only when their dimensionality is relevant, such as in the action-conditioned CPC objective. Executing the joint action $a_t^{1:N}$ through the unknown kernel $P$ yields the next state $s_{t+1}$ and the shared reward $r_t$. An episode ends when one army is eliminated or a time limit is reached. The goal is to maximize the discounted return $\mathbb{E}[\sum_{t=0}^{\infty}\gamma^t r_t]$ while respecting decentralized execution. We use discount factor $\gamma \in (0,1)$ for return computation, and a continuation indicator $c_t$ for episode termination. An overall view of our method is described in Figure 2.

**Design choices relative to prior work.** To make clear how MACT relates to existing multi-agent world-model baselines, Table 1 summarizes key architectural and training choices. MACT is closest in overall pipeline to tokenized per-agent world models with Perceiver-style cross-agent context, but differs in its prediction objective: we introduce an action-conditioned, multi-step contrastive loss that explicitly trains a $K_{\mathrm{cpc}}$-step predictive representation rather than only next-step reconstruction. We avoid joint cross-agent self-attention over all agents' tokens, which would scale quadratically with the effective sequence length, and instead use a single Perceiver-style cross-attention step that aggregates the current-step joint token set of length $N(K_{\mathrm{tok}}+1)$ into $N$ agent-wise global features. The formal complexity is given in Appendix B.

**Vector-to-token conversion.** First, MACT converts each agent's continuous observation vector into a structured sequence of discrete tokens. We use a small vector-quantized auto-encoder $(E, D, \mathcal{Z})$. This process creates a learned vocabulary for the features of the environment. The encoder takes the full observation vector $o_t^i$, splits it into $K_{\mathrm{tok}} = 8$ smaller pieces, and for each piece finds the nearest code-book entry $x_{t,j}^i \in \{1{:}256\}$:

$$x_t^i = (x_{t,1}^i, \ldots, x_{t,K_{\mathrm{tok}}}^i) \quad \text{with} \quad x_t^i \in \mathcal{Z}^{K_{\mathrm{tok}}}. \tag{1}$$

Throughout the paper, each $x_{t,j}^i \in \mathcal{Z}$ is a discrete code-book index. We denote the tokenizer code-book as $\mathcal{Z} = \{1, \ldots, 256\}$. Whenever a continuous representation is required, we explicitly apply the shared embedding in Eq. equation 2. That is, we write $e(x_{t,j}^i)$ for the corresponding $D_x$-dimensional embedding.

Using discrete tokens instead of raw continuous numbers brings two practical advantages. It stabilizes training because predicting the correct code from a fixed 256-entry vocabulary is a standard cross-entropy classification problem, avoiding the large gradients that can occur with direct regression. It also exposes useful compositional structure: treating observations as token sequences lets the model learn dependencies such as a token for small distance to an enemy often preceding a token for enemy in range. To form the input for one time step, the $K_{\mathrm{tok}}$ observation tokens are concatenated with a single discrete action token and a placeholder aggregation token $*_t^i$, yielding the step block $\mathbf{X}_t^i = \left[x_t^i, a_t^i, *_t^i\right]$. We count each action as one token in the sequence. Its one-hot encoding is only used in contexts where vector dimensionality matters explicitly, such as Eq. equation 5.

**Local Transformer dynamics.** The Transformer processes an agent's history and produces a summary of its current situation. To prepare the input, token blocks are taken for each agent $i$ individually from the start of the episode up to the current time $0{:}t$ and flattened into a long sequence. A block-sparse Transformer $\phi$ processes this sequence. The block-sparse design restricts self-attention so that each agent's Transformer

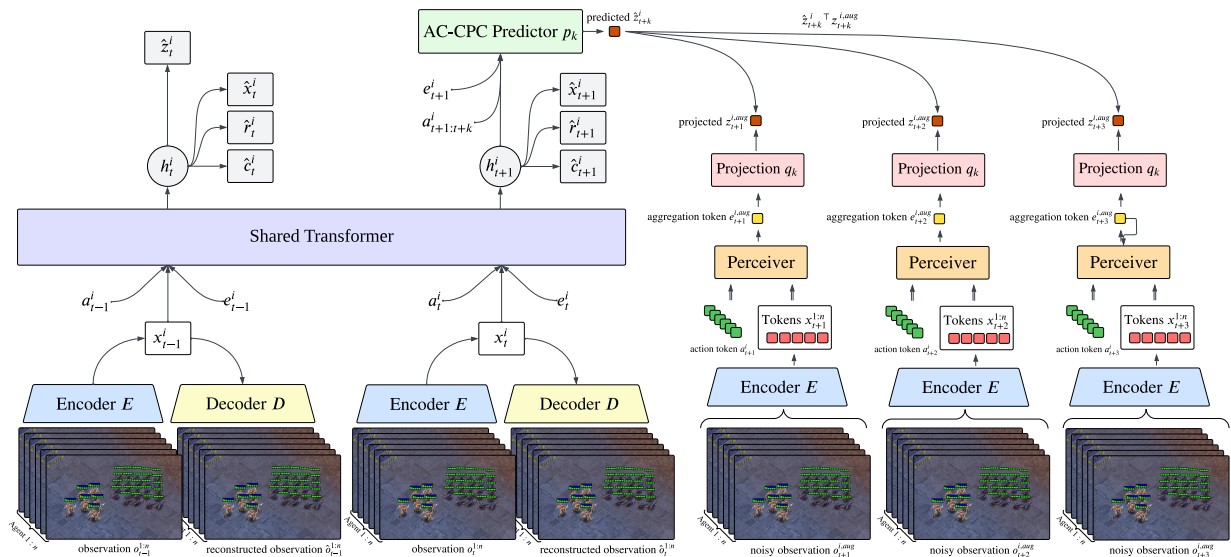

Figure 2: **World-model training in MACT.** Observations $o_t^{1:N}$ are tokenized by a VQ encoder $E$ into discrete codes $x_t^{1:N}$, and a decoder $D$ reconstructs $\hat{o}_t^{1:N}$ to train the tokenizer. Given token histories and executed actions, a shared per-agent Transformer produces local latents $h_t^i$, from which one-step prediction heads output $\hat{x}_{t+1}^i$, reward $\hat{r}_t^i$, and continuation $\hat{c}_t^i$. For AC-CPC, the predictor consumes the current representation $[h_t^i; e_t^i]$ together with a per-agent action window $a_{t:t+k-1}^i$ and predicts a future representation $\hat{z}_{t+k}^i$. This prediction is contrasted against the projected Perceiver target $z_{t+k}^{i,\text{aug}} = q_k(e_{t+k}^{i,\text{aug}})$, where $e_{t+k}^{i,\text{aug}}$ is computed from a dropout/noise-augmented future observation.

only attends to its own history at this stage, it cannot directly see the raw history of other agents. After reading its history, the model produces a single vector summarizing the current step. This vector is the local summary $h_t^i \in \mathbb{R}^{D_x}$, defined as the Transformer's output state at the aggregation-token position. Here $D_x$ is the token-embedding width.

**Centralized latent aggregation.** At each environment step $t$, we form a token set by concatenating all agents' observation tokens and the current discrete action token:

$$U_t \in \mathbb{R}^{N(K_{\text{tok}}+1) \times D_x}, \tag{2}$$

where $K_{\text{tok}}$ is the number of observation tokens per agent per step, and $D_x$ is the Transformer embedding width. We then apply a Perceiver-style cross-attention module that returns one latent per agent:

$$E_t = \text{LN}\Big(Q + \text{softmax}\big(\frac{(QW_Q)(U_t W_K)^\top}{\sqrt{d}}\big)(U_t W_V)\Big), \tag{3}$$

where $Q \in \mathbb{R}^{N \times D_e}$ are learned per-agent queries, $U_t \in \mathbb{R}^{N(K_{\text{tok}}+1) \times D_x}$ are current-step tokens, $W_Q \in \mathbb{R}^{D_e \times d}$, $W_K \in \mathbb{R}^{D_x \times d}$, and $W_V \in \mathbb{R}^{D_x \times D_e}$ are learned projections, $E_t = [e_t^1; \ldots; e_t^N] \in \mathbb{R}^{N \times D_e}$, and $d$ is the query/key dimension.

In summary, the Perceiver is a standard cross-attention block with learned queries, one per agent, and key/value projections, producing agent-wise latents by attending to the joint token set at the current time step. Thus, each global latent $e_t^i$ mixes information from all agents. Finally, the $e_t^i$ vectors are appended as extra tokens to the next step's Transformer input, propagating global context forward.

**Prediction heads and one-step losses.** The prediction heads attach directly to tokens produced by the local Transformer. The observation head reads the $k$-th latent slot, not $e_t^i$, and outputs a categorical distribution over the code-book to predict $\hat{x}_{t+1,k}^i$ conditioned on $x_{\leq t,\cdot}^i$, $a_{\leq t}^i$, $e_{\leq t}^i$, and the previously generated

slots $\hat{x}^i_{t+1,<k}$. This autoregressive factorization across the $K_{\text{tok}}$ slots captures intra-step structure such as the geometry of nearby units. The reward head maps the aggregation-slot hidden state $h^i_t$ through an MLP to produce a scalar reward prediction $\hat{r}^i_t$, and the discount head predicts the continuation indicator $\hat{c}^i_t \in (0,1)$. The one-step likelihood objective $\mathcal{L}_{\text{dyn}}$ averages token, reward, and continuation losses over agents and timesteps:

$$\mathcal{L}_{\text{dyn}} = \mathbb{E}_{i,t}\Big[ \sum_{k=1}^{K_{\text{tok}}} \text{CE}\big(\hat{x}^i_{t+1,k}, x^i_{t+1,k}\big) + \text{SmoothL1}\big(\hat{r}^i_t, \text{symlog}(r_t)\big) + \text{BCE}\big(\hat{c}^i_t, c_t\big)\Big]. \tag{4}$$

where the expectation denotes averaging over the sampled replay segment.

The previous components specify the latent world model: given tokenized observation-action histories, the shared Transformer plus the single Perceiver step produce per-agent local latents $h^i_t$ and global context features $e^i_t$. The objectives introduced next, $\mathcal{L}_{\text{dyn}}$, $\mathcal{L}_{\text{cpc}}$, and the auxiliary availability term, are applied to these latent quantities during world-model training. In the imagination-based actor–critic phase, we decode the world model's predicted observation tokens to reconstructed observations $\hat{o}^i_t$, which serve as inputs to the decentralized actors. The centralized critic aggregates information across agents under CTDE. The Perceiver outputs $e^i_t$ influence control through their effect on the learned latent dynamics.

**Action-conditioned contrastive prediction (per-agent).** Next-step supervision does not force the aggregation state $h^i_t$ to carry information about how this agent's planned actions will shape its future when teammates are also moving. MACT therefore asks each agent to predict, in latent space, what its Perceiver context will be several steps ahead given its own action window. Concretely, for $k \in \{0{:}K_{\text{cpc}}{-}1\}$ we form the context

$$\underbrace{h^i_t}_{\text{Transformer}} \ \| \ \underbrace{e^i_t}_{\text{Perceiver}} \ \| \ \underbrace{a^i_{t:t+k-1}}_{\text{Agent actions}} \quad (\in \mathbb{R}^{D_x + D_e + k\,|\mathcal{A}|}\,). \tag{5}$$

Here $h^i_t \in \mathbb{R}^{D_x}$ is the aggregation-slot hidden state of agent $i$ at time $t$, $e^i_t \in \mathbb{R}^{D_e}$ is its Perceiver-derived global context, and $a^i_{t:t+k-1} \in \{0,1\}^{k|\mathcal{A}|}$ is the concatenation of the next $k$ one-hot encodings $\text{onehot}(a^i_t), \dots, \text{onehot}(a^i_{t+k-1})$. Using per-agent actions preserves the identity of the maneuver being predicted, while $e^i_t$ already carries team context. The team-aggregated alternative is evaluated in Section 4.1.

A two-layer MLP $p_k \colon \mathbb{R}^{D_x + D_e + k|\mathcal{A}|} \to \mathbb{R}^{d_z}$ maps this concatenation to a projected prediction, and a two-layer projector $q_k \colon \mathbb{R}^{D_e} \to \mathbb{R}^{d_z}$ produces the projected target. Let $e^{i,\text{aug}}_{t+k}$ denote the Perceiver output computed from an augmented future observation view at time $t+k$. We then define

$$\hat{z}^i_{t+k} = p_k\Big(\big[h^i_t\|e^i_t\|a^i_{t:t+k-1}\big]\Big), \qquad z^i_{t+k} = q_k(e^{i,\text{aug}}_{t+k}). \tag{6}$$

The augmented view is used to reduce exact token matching and encourage action-relevant invariance, but, as the ablations show, its strength is map-dependent. With a minibatch of $Q$ positives (agent-time pairs), we index pairs by $q \in \{1, \dots, Q\}$. Let $Z_{t+k} = [z^{(1)}_{t+k}, \dots, z^{(Q)}_{t+k}] \in \mathbb{R}^{d_z \times Q}$. The InfoNCE term uses dot-product logits $\hat{z}^{(q)\top}_{t+k} Z_{t+k} \in \mathbb{R}^Q$ and cross-entropy over the index of the positive, the diagonal match:

$$\ell_k = \frac{1}{Q} \sum_{q=1}^{Q} \text{CE}\big(\hat{z}^{(q)\top}_{t+k} Z_{t+k}, q\big). \tag{7}$$

Intuitively, the predictor must learn a causal association: if agent $i$ executes $a^i_{t:t+k-1}$ while embedded in team context $e^i_t$, its future context should look like $z^i_{t+k}$. This resolves temporal ambiguity because the action window disambiguates which future is correct, mitigates credit assignment by linking an agent's actions to its own future state, and reduces non-stationarity because teammates' reactions are already absorbed into $e^i_t$. We include a same-step term ($k{=}0$) to align spaces and stabilize training, and weight farther horizons geometrically with $\lambda_{\text{cpc}}{=}0.75$:

$$\mathcal{L}_{\text{cpc}} = \sum_{k=0}^{K_{\text{cpc}}-1} \frac{\lambda^k_{\text{cpc}}}{\sum_{j=0}^{K_{\text{cpc}}-1} \lambda^j_{\text{cpc}}} \, \ell_k. \tag{8}$$

This keeps gradients from distant steps present without letting them dominate optimization.

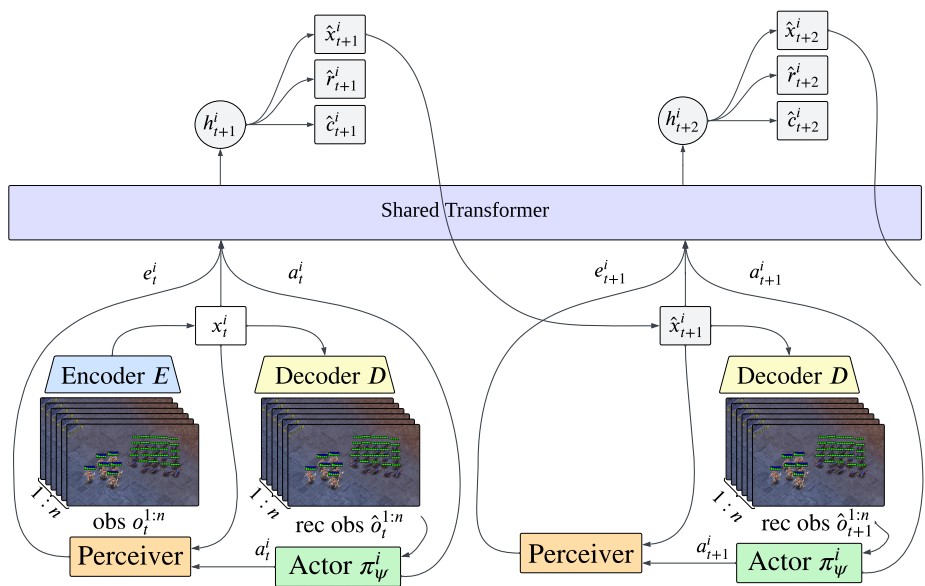

Figure 3: A replay state is encoded into tokens $x_t^{1:N}$ and reconstructed as $\hat{o}_t^{1:N}$. The frozen world model predicts $\hat{x}_{t+1}^{1:N}$, rewards $\hat{r}_t$, and continuation $\hat{c}_t$, which are decoded to $\hat{o}_{t+1}^{1:N}$ and fed back to the actors. Each agent samples actions from reconstructed inputs, $a_t^i \sim \pi_\psi^i(\hat{o}_t^i)$, unrolling for $H_{\text{roll}}$ steps without environment interaction.

**Available-action regularizer.** SMAC provides an available-action mask $m_t^i \in \{0,1\}^{|\mathcal{A}|}$ that disables infeasible actions. Without extra care, the logits of masked actions can drift to large negative values, which we found to hinder optimization. We add an auxiliary head that predicts the next-step availability mask $m_{t+1}^i$ from the agent latent, yielding $\hat{m}_{t+1}^i \in (0,1)^{|\mathcal{A}|}$. We train it with a per-action Bernoulli loss

$$\mathcal{L}_{\text{av}} = \mathbb{E}_{i,t}\Big[\text{BCE}\big(\hat{m}_{t+1}^i,\, m_{t+1}^i\big)\Big]. \tag{9}$$

We add this term to the world-model objective alongside the one-step likelihood and AC-CPC:

$$\mathcal{L}_{\text{WM}} = \mathcal{L}_{\text{dyn}} + \beta_{\text{cpc}}\mathcal{L}_{\text{cpc}} + \beta_{\text{av}}\mathcal{L}_{\text{av}}. \tag{10}$$

We found the natural scales of these terms to be comparable and therefore use equal weights ($\beta_{\text{cpc}} = \beta_{\text{av}} = 1$). Training details and hyperparameters are deferred to Appendix B.

**Imagination-based actor–critic.** After every world-model update we keep its parameters fixed and perform imagination rollouts in latent space. Given an initial real observation from the replay buffer, we encode it once with the tokenizer and world model to obtain the per-agent local latents $h_0^{1:N}$. We then unroll $H_{\text{roll}} = H$ latent steps without accessing the environment.

At each imagined step $t$, every agent runs a decentralized actor $\pi_\psi^i : \mathbb{R}^{d_o} \to \mathbb{R}^{|\mathcal{A}|}$ that takes the reconstructed observation $\hat{o}_t^i$ as input and outputs action logits. We obtain $\hat{o}_t^i$ by decoding the world model's predicted observation tokens, i.e., $\hat{o}_t^i = D(\hat{x}_t^i)$. The actor is implemented as a two-layer MLP with hidden width 256 and ELU activations. Infeasible actions are masked using the SMAC available-action vector, and an action is sampled from the resulting categorical distribution. The frozen world model then uses these imagined actions to predict synthetic rewards, continuation flags, and the next-step observation tokens $\hat{x}_{t+1}^{1:N}$, which are decoded to $\hat{o}_{t+1}^{1:N}$ for the next actor step.

For value estimation we employ a centralized critic $V_\omega : \mathbb{R}^{ND_x} \to \mathbb{R}$ that aggregates information across agents by concatenating the local latents $\{h_t^1, \dots, h_t^N\}$ and passing them through a two-layer MLP with hidden width 256 and ELU activations. The resulting scalar output approximates the value of the imagined latent

state. Using the synthetic rewards and continuation flags, we compute $\lambda$-returns along the latent trajectory and update the actors by advantage policy gradient with an entropy bonus, and the critic by a symlog mean-squared error loss. During this phase gradients do not flow into the world model, ensuring that policy learning does not destabilize the learned dynamics.

## 4 Experiments

We evaluate on SMAC under a low-data protocol. All methods are re-run in a unified harness with a pinned SC2.4.1.2.60604 build, identical per-map environment-step budgets, and the same 100-episode greedy evaluation. The main 50K-step set contains ten maps used by prior world-model work. We use 200K steps for the harder `3s_vs_5z` map and for two SuperHard maps, `corridor` and `6h_vs_8z`. Baselines are MARIE (Zhang et al., 2025), MATWM (Deihim et al., 2025), MAMBA (Egorov & Shpilman, 2022), MBVD (Xu et al., 2022), MAPPO (Yu et al., 2022), and MAT (Wen et al., 2022). Appendix B lists rollout budgets and hyperparameters for all methods.

Table 2: SMAC comparison under the unified protocol. Cells report mean±std of win rate (%), smoothed over the last 5 evaluations. Dark gray stands for best result, and light gray stands for second.

| Map | Steps | MACT (Ours) | MARIE (2024) | MATWM (2025) | MAMBA (2022) | MBVD (2022) | MAPPO (2022) | MAT (2022) |
|---|---|---|---|---|---|---|---|---|
| 2m_vs_1z | 50K | 99.2±1.2 | 52.0±44.9 | 58.1±13.2 | 92.0±11.7 | 0.0±0.0 | 56.8±16.6 | 81.0±20.4 |
| 2s_vs_1sc | 50K | 91.9±7.4 | 92.0±7.5 | 96.7±3.1 | 86.0±13.9 | 0.0±0.0 | 60.0±11.4 | 23.9±13.0 |
| 2s3z | 50K | 35.6±17.4 | 25.0±11.0 | 43.9±16.5 | 46.0±12.4 | 65.0±11.5 | 0.9±0.8 | 0.4±0.5 |
| 3m | 50K | 91.2±6.2 | 96.5±3.7 | 79.3±7.7 | 92.0±6.0 | 43.0±20.6 | 57.4±11.0 | 42.0±3.5 |
| 3s_vs_3z | 50K | 82.5±4.7 | 85.0±16.4 | 23.2±25.3 | 76.0±18.0 | 0.0±0.0 | 3.6±3.5 | 1.1±0.7 |
| 3s_vs_4z | 50K | 3.8±5.3 | 9.0±11.1 | 0.1±0.2 | 5.0±10.0 | 0.0±0.0 | 0.0±0.0 | 0.0±0.0 |
| 8m | 50K | 95.0±5.0 | 48.5±23.2 | 0.0±0.0 | 65.0±12.6 | 74.1±13.6 | 19.5±5.7 | 28.2±8.9 |
| MMM | 50K | 43.3±7.2 | 23.5±10.7 | 2.8±4.1 | 60.0±15.8 | 9.0±3.9 | 0.8±1.5 | 0.5±1.0 |
| so_many_baneling | 50K | 94.2±2.4 | 56.5±26.6 | 0.0±0.0 | 77.0±16.0 | 11.2±3.1 | 33.5±9.5 | 32.9±4.2 |
| 3s_vs_5z | 200K | 95.0±3.5 | 66.5±16.4 | 0.0±0.0 | 15.0±13.4 | 0.0±0.0 | 0.0±0.0 | 0.0±0.0 |
| corridor | 200K | 77.5±20.9 | 38.8±31.2 | 0.0±0.0 | 10.0±14.1 | 0.0±0.0 | 0.0±0.0 | 0.0±0.0 |
| 6h_vs_8z | 200K | 0.0±0.0 | 0.0±0.0 | 0.0±0.0 | 0.0±0.0 | 0.0±0.0 | 0.0±0.0 | 0.0±0.0 |
| Mean | — | 67.4 | 49.4 | 25.3 | 52.0 | 16.9 | 19.4 | 17.5 |
| Median | — | 86.9 | 50.2 | 1.4 | 62.5 | 0.0 | 2.2 | 0.8 |

Table 2 shows a different picture from the original cross-paper comparison. Under a pinned and unified protocol, MATWM does not reproduce its previously published strength and is no longer the closest competitor. MAMBA is the strongest baseline in aggregate, and it is best on `MMM`. MACT nevertheless has the highest mean and median across the 12-map table, leads on five maps, and performs especially well on `2m_vs_1z`, `8m`, `so_many_baneling`, `3s_vs_5z`, and `corridor` SuperHard map. The results also expose clear limitations: `3s_vs_4z` remains low for every method, `2s3z` favors MBVD/MAMBA, and `6h_vs_8z` is unsolved by all methods at this budget.

Because individual SMAC maps can be high variance, we report aggregate uncertainty using the rliable protocol (Agarwal et al., 2021) on all 12 evaluated maps. MACT reaches an IQM win rate of 80.4, compared with 60.9 for MAMBA, 52.1 for MARIE, 14.1 for MATWM, 11.3 for MAPPO, 10.0 for MAT, and 3.8 for MBVD. The bootstrap intervals at Figure6 in Appendix A show that MACT is separated from every baseline. The probability of improvement is 0.68 versus MARIE, 0.63 versus MAMBA, 0.81 versus MATWM, 0.93 versus MAPPO, 0.92 versus MAT, and 0.85 versus MBVD. Appendix A provides the corresponding learning curves, performance profiles, and rollout-fidelity plots.

### 4.1 Ablation Studies

We conduct ablation studies to isolate the impact of MACT's key components. All ablations keep the architecture, optimizer, and implementation protocol fixed to Table 3. Where only the factor under test is

varied. Figure 4 contains three experiments motivated by reviewer feedback: the CPC horizon study includes the requested one-step control $K_{\mathrm{cpc}} = 1$, the augmentation study is moved to the harder heterogeneous `MMM` map, and the conditioning study compares per-agent action windows with a team-aggregated action summary.

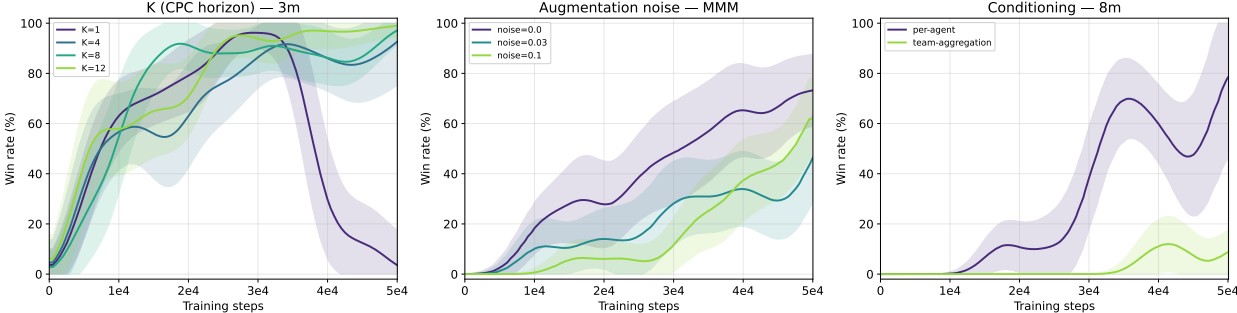

Figure 4: Ablations under the unified implementation. Left: CPC horizon on `3m`, including the one-step $K_{\mathrm{cpc}} = 1$ control. Middle: positive-target augmentation noise on the harder `MMM` map. Right: per-agent action conditioning versus a team-aggregated action summary on `8m`.

**CPC horizon.**   We vary the action-conditioned CPC horizon on `3m`. The one-step control $K_{\mathrm{cpc}} = 1$ initially learns but later collapses, indicating that a degenerate one-step contrastive head is not sufficient to explain MACT's gains. In contrast, $K = 4, 8, 12$ remain stable. $K = 4$ learns quickly at the beginning but provides less temporal coverage. While $K = 8$ gives the best trade-off between fast learning and a high plateau, and $K = 12$ can approach similar performance but with more variance, likely because the predictor must consume longer action windows and align more distant positives. Overall, the ablation supports the intended multi-step design: the objective is most useful when it predicts beyond the immediate next transition without making the contrastive task unnecessarily long.

**Augmented positive targets.**   We test feature dropout/noise applied only when forming CPC positives on `MMM`. On this harder heterogeneous-unit map, no augmentation reaches the highest final win rate. $p = 0.03$ lags, and $p = 0.1$ improves late but remains below the no-noise curve at 50K steps. Thus, the role of augmentation is not a universal performance boost. It can make the positive target less vulnerable to exact token matching, but its strength should be treated as a map-dependent regularizer. We therefore use the ablation to narrow the claim rather than to argue that light dropout is always beneficial.

**Per-agent vs. team aggregated conditioning.**   For completeness we test an agent-agnostic variant that replaces own actions with an aggregated joint-action summary at each step:

$$\tilde{a}_{t:t+k-1}^{i} \; = \; \mathrm{Agg}\!\left(a_{t:t+k-1}^{1:N}\right).$$

To keep inputs comparable, the query features are also aggregated to team level ($h_t^{\mathrm{team}} = \frac{1}{N}\sum_i h_t^i$, $e_t^{\mathrm{team}} = \mathrm{Agg}(e_t^{1:N})$) before applying the same context construction and CPC loss.

On `8m`, conditioning on each agent's own future actions clearly outperforms using an aggregated joint-action summary. The per-agent curve lifts off earlier and reaches a much higher final win rate, while the team-aggregated variant remains near zero for most of training and finishes far lower. We attribute this gap to identity-agnostic pooling blurring who executed which maneuver. AC-CPC learns best from a precise mapping between an agent's action window and the evolution of its own Perceiver latent, while global team context is already provided by $e_t^i$. Adding a coarse joint-action summary appears redundant and noisier, increasing predictor input dimensionality and weakening contrastive alignment. We therefore adopt per-agent conditioning as the default conditioning mechanism in MACT.

## 5 Limitations

MACT is evaluated in the low-data SMAC regime, which isolates the sample-efficiency and coordination challenges that motivate action-conditioned multi-step prediction. The claims are therefore scoped to structured observations, discrete actions, and tight interaction budgets. The SuperHard maps broaden the difficulty range, but `6h_vs_8z` remains unsolved by all methods at 200K steps. Extending MACT to GRF or MA-MuJoCo is important future work, but it is not a drop-in re-run: visual or continuous-control settings require a different observation encoder/tokenizer and a continuous-action conditioning representation. The ablations also reveal map-dependent sensitivities, especially for augmentation and micro-intensive maps, so we avoid claiming universal hyperparameter robustness.

## 6 Conclusion

MACT is a decentralized action-conditioned Transformer world model for cooperative MARL. It combines per-agent token dynamics, centralized Perceiver context, and an AC-CPC objective that aligns planned actions with multi-step latent dynamics. Under a unified SMAC evaluation, MACT achieves the strongest aggregate IQM and the best mean/median table performance while making uncertainty and failure cases explicit. The ablations show that multi-step horizons and per-agent action conditioning are central, while augmentation should be treated as a map-dependent regularizer. Rollout-fidelity diagnostics further support the representation-learning motivation behind AC-CPC: MACT's observation predictions drift more slowly over longer imagined horizons.

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

# A    Appendix: Training results

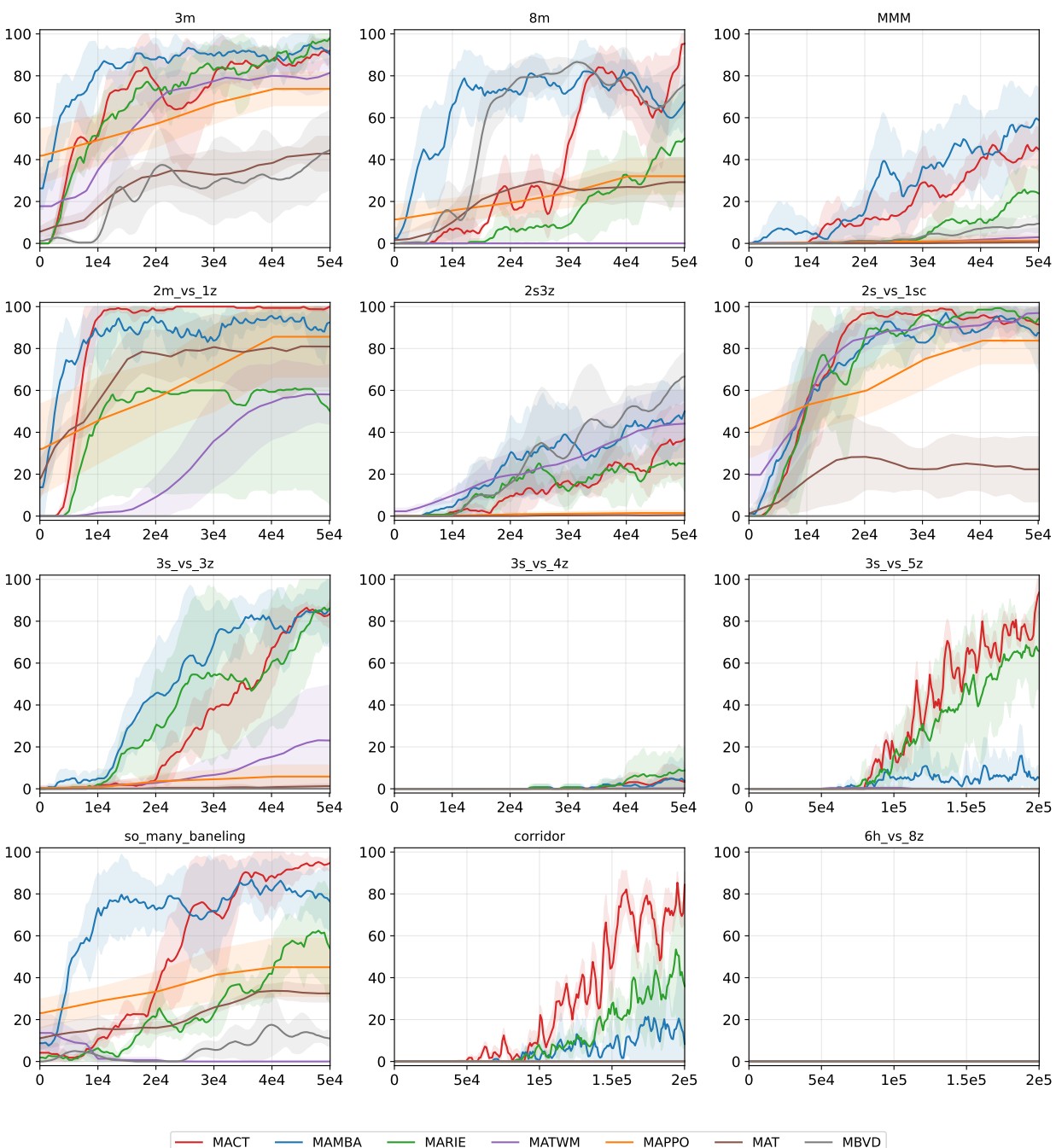

Figure 5: Evaluation win rate vs. environment steps for *all* methods, re-run under the unified protocol (mean±std over seeds, Gaussian-smoothed). All methods share the same per-map step budget, so curves are directly comparable in both final performance and sample efficiency.

# B    Appendix: MACT and baselines implementation

**Reproducibility and setup.**    All ablations are run on the same implementation as our main method and keep the shared world-model training setup fixed wherever applicable. To minimize environment drift, we

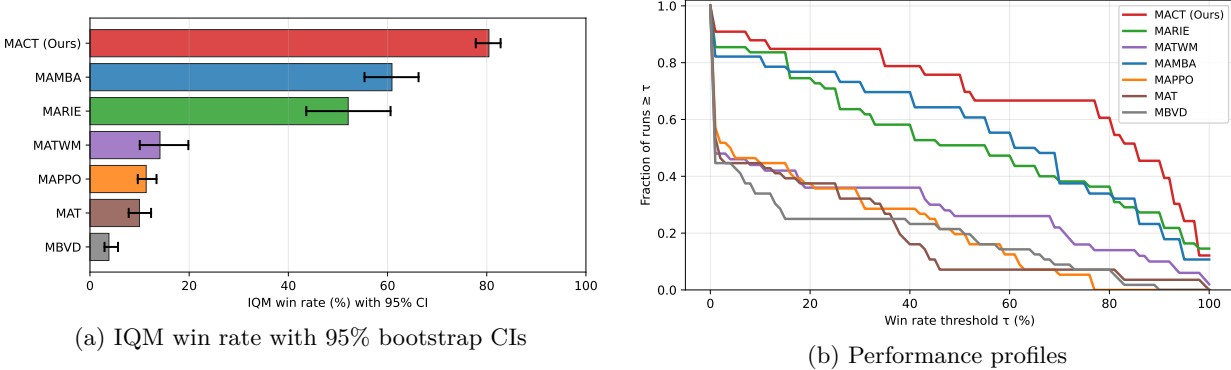

(a) IQM win rate with 95% bootstrap CIs

(b) Performance profiles

Figure 6: Rigorous evaluation (rliable-style) over all 12 evaluated SMAC maps using per-seed scores. (a) IQM with stratified-bootstrap 95% CIs: MACT achieves the highest aggregate score. (b) Fraction of runs exceeding a win-rate threshold $\tau$. Probability of improvement $P(\text{MACT} > X)$: MARIE 0.68, MAMBA 0.63, MATWM 0.81, MAPPO 0.93, MAT 0.92, MBVD 0.85.

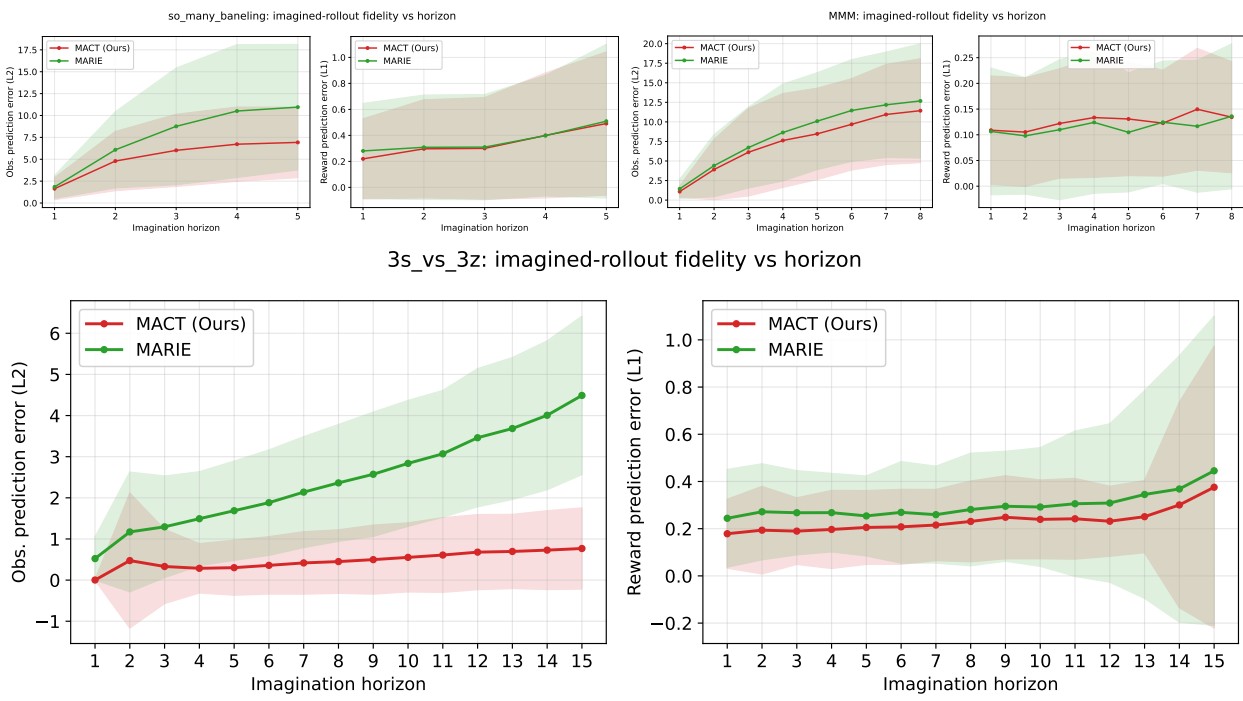

Figure 7: Imagined-rollout fidelity vs. horizon (teacher-force, identical ground-truth actions fed to both world models), on maps spanning imagination horizons $H$=5/8/15. Left panels: observation prediction error (L2). Right panels: reward error (L1). MACT's observation error grows markedly slower with horizon (e.g. on `3s_vs_3z`, $H$=15: MACT $\leq$ 0.8 vs. MARIE 4.5, $\sim 6\times$), while reward error is comparable. Supporting the claim of more consistent multi-step *state* representations.

pin the StarCraft II client to *SC2.4.1.2.60604* (the same version used by MATWM). Different SC2 builds can yield different win rates, which is why the unified re-run is important. Our codebase targets Python 3.11 and PyTorch 2.x. Exact package versions, training scripts, per-map configs, the CSV files used to render plots, and the plotting scripts themselves will be made publicly available upon completion of the review process.

**Hyperparameters.** We keep global training knobs identical across main and ablation runs (optimizer families, learning rates, clipping, entropy coefficient, etc.. See Table 5). For map-dependent imagination settings, we use $H{=}15$ and 4 policy-update epochs for small-agent maps such as `2m_vs_1z`, `3m`, and `3s_vs_5z`. $H{=}8$ and 10 epochs for larger maps such as `8m`, `MMM`, `corridor`, and `6h_vs_8z`. While the explicit $H{=}5$, 30-epoch setting for `2s3z` and `so_many_baneling`. Unless stated otherwise, the tokenizer and world model are each trained for 200 epochs, we perform 5 PPO epochs per policy update, and the $\lambda$-return uses $\lambda{=}0.95$ with $\gamma{=}0.99$. Map-specific overrides (e.g., $K_{\mathrm{cpc}}$) follow Table 5.

Table 3: Full hyperparameters for the world-model baselines. MACT uses the tokenized Transformer/Perceiver world-model settings plus the AC-CPC latent-augmentation objective. Dashes denote settings that do not apply to a given method.

| Hyperparameter | **MACT (Ours)** | MARIE | MAMBA | MATWM |
|---|---|---|---|---|
| *World model* | | | | |
| Tokenizer / latent | VQ (vocab 512) | VQ (vocab 512) | RSSM (cont.) | VQ-categorical |
| Token embed. dim | 128 | 128 | — | — |
| Transformer embed. dim | 256 | 256 | — | 512 |
| WM Transformer depth / layers$^\S$ | 10 | 10 | — | 2 |
| Perceiver depth (cross-agent)$^\S$ | 2 | 2 | — | — |
| Transformer context length | $H$ | $H$ | — | 64 |
| WM learning rate | $1{\times}10^{-4}$ | $1{\times}10^{-4}$ | $2{\times}10^{-4}$ | $3{\times}10^{-5}$ |
| WM weight decay | 0.01 | 0.01 | — | — |
| WM batch size | 30 | 30 | 40 | 64 |
| WM updates per round | 200 | 200 | 60 | 1/step |
| Tokenizer batch / lr | 256 / $3{\times}10^{-4}$ | 256 / $3{\times}10^{-4}$ | — | — |
| *Actor–critic / policy* | | | | |
| Imagination horizon $H$ | 5/8/15 | 5/8/15 | 15 | 16 |
| AC updates per round$^\P$ | 30/10/4 | 30/10/4 | 4 | 1/step |
| PPO epochs | 5 | 5 | 5 | — |
| AC batch (imag. traj.) | 600 | 600 | 40 | 512 |
| Actor / Value lr | $5{\times}10^{-4}$ | $5{\times}10^{-4}$ | $5{\times}10^{-4}$ | $3{\times}10^{-4}$ |
| Discount $\gamma$ | 0.99 | 0.99 | 0.99 | 0.985 |
| GAE / $\lambda$-return | 0.95 | 0.95 | — | 0.95 |
| Entropy coef. | 0.001* | 0.001* | 0.001 | 0.001 |
| Hidden size | 256 | 256 | 256 | 512 |
| Target update interval | 20 | 20 | 1 | EMA |
| Grad clip / max-norm | 100 / 10 | 100 / 10 | 100 | — |
| *Replay / collection* | | | | |
| Buffer capacity | 250k | 250k | 250k | 50k |
| Min buffer before training | 1000 | 1000 | 500 | 1000 |
| Collect interval (env steps) | 100$^\ddagger$ | 100$^\ddagger$ | 1 | 1 |
| *AC-CPC (MACT only)* | | | | |
| CPC horizon $K$ | 5 or 8$^\dagger$ | — | — | — |
| Latent-aug. noise scale | 0.1 | — | — | — |
| CPC temperature | 1.0 | — | — | — |
| CPC conditioning | per-agent | — | — | — |

\* Raised to 0.01 on 2m_vs_1z (and on 2s3z for MACT's raised-entropy variant). $^\dagger$ $K{=}5$ on so_many_baneling and 2s3z, $K{=}8$ otherwise. $^\ddagger$ `N_SAMPLES` ($= 200$ on 3s_vs_5z).

**Computational complexity.** Let $N$ be the number of agents, $H$ the sampled segment length, $K_{\mathrm{cpc}}$ the CPC horizon, $Q$ the number of (time,agent) positives in the minibatch, and $d_z$ the projection width. Because MACT keeps the MARIE-style tokenized Transformer/Perceiver backbone unchanged, it inherits

Table 4: Full hyperparameters for the model-free (MAPPO, MAT) and model-based value-decomposition (MBVD) baselines.

| Hyperparameter | MAPPO | MAT | MBVD |
|---|---|---|---|
| Backbone | RNN actor-critic | Transformer policy | QMIX + latent imag. |
| Episode length | 400 | 100 | — |
| Rollout threads | 1 | 1 | 1 |
| PPO epochs | 15 | 15 | — |
| Mini-batches | 1 | 1 | — |
| Clip $\epsilon$ | 0.2* | 0.05 | — |
| Entropy coef. | 0.01 | 0.01 | — |
| Actor lr | $5\times10^{-4}$ | $5\times10^{-4}$ | $5\times10^{-4}$ |
| Critic lr | $5\times10^{-4}$ | — | $5\times10^{-4}$ |
| Discount $\gamma$ | 0.99 | 0.99 | 0.99 |
| GAE $\lambda$ | 0.95 | 0.95 | — |
| Hidden size | 64 | 64 (embd) | 64 (rnn) |
| Layers / blocks | 1 | 1 block, 1 head | — |
| Imag. rollout depth | — | — | 3 |
| Mixing embed dim | — | — | 32 |
| Hypernet (layers/embed) | — | — | 2 / 64 |
| Batch / buffer (episodes) | — | — | 32 / 32 |

\* MAPPO uses recurrent MAPPO (rmappo) by default. On 3s_vs_4z and 3s_vs_5z it uses non-recurrent MAPPO with 4 stacked frames, and clip $\epsilon$=0.05 on 3s_vs_5z.

Table 5: Per-environment hyperparameter overrides for MACT and the tokenized world-model baseline. Imagination horizon $H$, imagination-update count (EPOCHS), and CPC horizon $K$ follow the same agent-count schedule across maps. All other values default to Table 3. Map-specific notes list the remaining deviations. $K$ applies to MACT only.

| Map | $n$ agents | $H$ | EPOCHS | $K_{\text{CPC}}$ | Budget | Map-specific overrides |
|---|---|---|---|---|---|---|
| 3m | 3 | 15 | 4 | 8 | 50k | — |
| 8m | 8 | 8 | 10 | 8 | 50k | — |
| MMM | 10 | 8 | 10 | 8 | 50k | WM/AC batch $\to$ 20 ($\geq$ 9 agents) |
| 2m_vs_1z | 2 | 15 | 4 | 8 | 50k | entropy 0.01, distributional reward loss (ce_for_r) |
| 2s3z | 5 | 5 | 30 | 5 | 50k | raised-entropy variant: entropy 0.01 |
| 2s_vs_1sc | 2 | 15 | 4 | 8 | 50k | — |
| 3s_vs_3z | 3 | 15 | 4 | 8 | 50k | — |
| 3s_vs_4z | 3 | 15 | 4 | 8 | 50k | — |
| 3s_vs_5z | 3 | 15 | 4 | 8 | 200k | sampling temperature 2.0, N_SAMPLES= 200 |
| so_many_baneling | (expl.) | 5 | 30 | 5 | 50k | explicit $H$=5 set |
| corridor | 6 | 8 | 10 | 8 | 200k | — |
| 6h_vs_8z | 6 | 8 | 10 | 8 | 200k | — |

the same near-linear scaling in team size: per-agent local Transformer dynamics avoid joint cross-agent self-attention over all tokens, and a single Perceiver-style cross-attention step operates over the current-step token set of length $L = N(K_{\text{tok}}+1)$. Relative to this one-step backbone without AC-CPC, the additional cost comes only from the contrastive heads and their in-batch similarity computation. For each $k \in \{0, \ldots, K_{\text{cpc}} - 1\}$, MACT applies per-agent MLP projections $p_k(\cdot)$ and $q_k(\cdot)$, giving an added projection cost $O\left(NH \sum_{k=0}^{K_{\text{cpc}}-1} C_{\text{MLP}}(k)\right)$, where $C_{\text{MLP}}(k)$ depends on the input width $D_x + D_e + k|\mathcal{A}|$. The InfoNCE term computes a batched similarity between $Q$ queries and $Q$ keys, implemented as a dense matrix multiply this costs $O(Q^2 d_z)$ per $k$. Overall, AC-CPC adds an approximately linear-in-$NHK_{\text{cpc}}$ projection term plus a total $O(K_{\text{cpc}} Q^2 d_z)$ similarity term, while leaving the adopted backbone architecture unchanged. In practice, $Q$ is the sampled minibatch size of agent–time pairs used for InfoNCE. With this sampling budget fixed, the added similarity cost does not change the backbone's scaling behavior with $N$. If all agent–time pairs were contrasted, the term should instead be read as scaling through $Q$.

Table 6: Empirical compute overhead of MACT over the comparable one-step world-model baseline, measured as a controlled per-world-model-update micro-benchmark (median over 20 iterations on a RTX 6000 Pro running an identical synthetic batch). AC-CPC lives only in the world-model loss, so the world-model update is the entire compute difference. The actor–critic update is identical for both methods. The relative time overhead shrinks as the map (and thus the base world model) grows.

| Map | $n$ | WM update (ms) | | | Peak GPU mem (GB) | | |
| --- | --- | --- | --- | --- | --- | --- | --- |
| | | MARIE | MACT | $\Delta$ | MARIE | MACT | $\Delta$ |
| 3m | 3 | 333 | 583 | +75% | 7.53 | 8.14 | +8.1% |
| MMM | 10 | 469 | 593 | +26% | 7.41 | 8.07 | +8.9% |

**Pseudocode overview.** Algorithm 1 summarizes MACT in three phases. (i) *Experience collection* gathers short trajectories in SMAC and stores $\{o, a, r, \gamma\}$ in a replay buffer. (ii) *World-model update* samples a segment, builds per-step team context with one Perceiver cross-attention to obtain $e_t^{1:N}$, and runs the shared per-agent Transformer to produce $h_t^i$. The one-step likelihood (token, reward, discount, and optional availability heads) is combined with a *per-agent* action-conditioned CPC loss: for $k=0{:}K_{\text{cpc}}-1$, a predictor consumes $\left[h_t^i; e_t^i; a_{t:t+k-1}^i\right]$ (for $k=0$ the action window is empty) and is trained via InfoNCE against a *projected* Perceiver latent from an augmented future view. While terms are weighted geometrically by $\lambda_{\text{cpc}}$. (iii) *Policy learning in imagination (CTDE)* freezes the world model and unrolls $H_{\text{roll}}$ latent steps. Masked decentralized actors produce actions, a centralized critic evaluates returns, and $\lambda$-returns train the agents.

---

**Algorithm 1: MACT** training pseudocode (per-agent)

---

**Input** : Replay buffer $\mathcal{D}$;
horizons $H$ (model), $H_{\mathrm{roll}}$ (imagination);
CPC horizon $K_{\mathrm{cpc}}$;
decay $\lambda_{\mathrm{cpc}}$
**Modules:** VQ tokenizer encoder $E$;
shared per-agent Transformer $\phi$;
Perceiver cross-attn $\theta$;
heads: obs-token, reward, discount, avail;
CPC heads $\{p_k, q_k\}$;
actors $\{\pi_\psi^i\}$;
critic $V_\xi$

**for** *epoch* $= 1, 2, \ldots$ **do**
    $o \leftarrow \text{ENV.RESET}()$          // (i) Collect real experience
    **repeat**
        sample $a_t^i \sim \pi_\psi^i(o_t^i)$ (mask infeasible)
        $(o', r, \text{done}, m) \leftarrow \text{ENV.STEP}(a^{1:N})$      // $m_t$ is avail mask
        push $\{o, a, r, \text{done}, m\}$ to $\mathcal{D}$;
        $o \leftarrow \text{done? } \text{ENV.RESET}() : o'$
    **until** *n steps*
    sample segment $\{o_t^{1:N}, a_t^{1:N}, r_t, c_t, m_t\}_{t=\tau}^{\tau+H-1} \sim \mathcal{D}$      // (ii) Train world model
    $x_t^i \leftarrow E(o_t^i)$ for all $t, i$      // vector $\to$ discrete tokens
    **for** $t = \tau$ **to** $\tau + H - 1$ **do**
        $e_t^{1:N} \leftarrow \text{PERCEIVER}_\theta(\{x_t^i\}, a_t^{1:N})$
        $h_t^i \leftarrow \text{TRANSFORMER}_\phi([x_{\leq t}^i, a_{\leq t}^i, e_{\leq t}^i])$
        **for** $j = 1$ **to** $K_{tok}$ **do**
            predict $\hat{x}_{t+1,j}^i$ autoregressively conditioning on $\hat{x}_{t+1,<j}^i$
        predict $\hat{r}_t^i, \hat{c}_t^i, \hat{m}_{t+1}^i$ from $h_t^i$
    build boundary mask $\mathcal{M}$ on flattened streams $y_{1:T}^i$
    $\mathcal{L}_{\mathrm{dyn}} \leftarrow \mathcal{L}_{\mathrm{token}}(\mathcal{M}) + \mathcal{L}_{\mathrm{reward}} + \mathcal{L}_{\mathrm{discount}} + \mathcal{L}_{\mathrm{av}}$
    $o_t' \leftarrow \text{AUGMENT\_VECTOR}(o_t)$;      // Per-agent AC-CPC (augmented future Perceiver latents)
    $x_t' \leftarrow E(o_t')$
    $e_t'^{1:N} \leftarrow \text{PERCEIVER}_\theta(\{x_t'^i\}, a_t^{1:N})$ for all $t$
    **for** $k = 0$ **to** $K_{cpc}-1$ **do**
        **foreach** $(i, t)$ *in minibatch* **do**
            $\hat{z}_{t+k}^i \leftarrow p_k([h_t^i; e_t^i; a_{t:t+k-1}^i])$
            $z_{t+k}^i \leftarrow q_k(e_{t+k}'^i)$
        compute InfoNCE $\ell_k$ with in-batch negatives
    $\mathcal{L}_{\mathrm{cpc}} \leftarrow \sum_{k=0}^{K_{\mathrm{cpc}}-1} \dfrac{\lambda_{\mathrm{cpc}}^k}{\sum_{j=0}^{K_{\mathrm{cpc}}-1} \lambda_{\mathrm{cpc}}^j} \ell_k$
    **Update** $(E, \phi, \theta, \{p_k, q_k\})$ on $\mathcal{L}_{\mathrm{WM}} = \mathcal{L}_{\mathrm{dyn}} + \mathcal{L}_{\mathrm{cpc}}$
    init $(o_0, a_0) \sim \mathcal{D}$;
    $x_0^i \leftarrow E(o_0^i)$;      // (iii) Train actors in imagination (CTDE)
    $e_0^{1:N} \leftarrow \text{PERCEIVER}_\theta(\{x_0^i\}, a_0^{1:N})$;
    $h_0^i \leftarrow \text{TRANSFORMER}_\phi([x_{\leq 0}^i, a_{\leq 0}^i, e_{\leq 0}^i])$
    **for** $t = 0$ **to** $H_{roll} - 1$ **do**
        sample $a_t^i \sim \pi_\psi^i(\hat{o}_t^i)$ (mask infeasible)
        world model predicts $\hat{r}_t, \hat{c}_t$, and autoregressively predicts $\hat{x}_{t+1}^i$
        $\hat{o}_{t+1}^i \leftarrow D(\hat{x}_{t+1}^i)$      // decode predicted tokens
        update $(e_{t+1}^{1:N}, h_{t+1}^{1:N})$ from $(\hat{x}_{t+1}^{1:N}, a_{t+1}^{1:N})$
    update $\psi$ by advantage PG and $\xi$ by symlog MSE

---

