# OpenReview forum: "Action-Conditioned Transformers for Decentralized Multi-Agent World Models"
_TMLR — Decision pending for TMLR_

### Review · Reviewer_pA2w · 2026-05-27

**Summary Of Contributions:**

This paper proposes a model-based multi-agent world model method specifically designed for low-data regime environments. The authors test it out on various SC2 maps and the experiments show that the proposed method beats out the current baselines.

Strengths

_Effectively tackles the challenge of running MARL in low-data environments using a world model.
_Shows solid empirical superiority on complex StarCraft II maps.

Weaknesses

_The baseline comparisons might not be perfectly apples-to-apples regarding codebases, seeds, or SC2 versions.
_Learning curves of evaluation win rates are only provided for the proposed method in the ablation study, missing comparisons with the baselines.
_Lacks a comparison against fairer, transformer-based or sequence-modeling baselines.

**Additional Comments:**

N/A

**Audience:**

Yes

**Audience Explanation:**

Figuring out how to make multi-agent reinforcement learning (MARL) work well in low-data regimes, and applying the world model framework is a promising approach that is interesting for MARL researchers.

**Claims And Evidence:**

Yes

**Claims Explanation:**

While the results on the SC2 maps look good and show the method's superiority, the evidence isn't fully convincing yet. Right now, MAPPO is the only model-free baseline used. To really back up the claims, the paper needs to compare against transformer-based sequence modeling methods.

Authors mention that they report the win-rate numbers for baselines from the respective papers, it needs to be completely clear that all baselines were run under the exact same conditions (codebase, version, seeds, etc.).

Learning curves of evaluation win-rates are only provided for the proposed method in the ablation study. We need to see evaluation return curves for both the proposed method and the baselines to properly evaluate the performance gains and sample efficiency.

**Requested Changes:**

Experiments:

_You should rerun all baselines under the same codebase, same SC2 version, same seeds, same evaluation protocol, and same data budget.

_The only model-free baseline used in the experiments is MAPPO. You need to compare this to transformer-based baselines such as Multi-Agent Transformer (Wen et al., 2022, "Multi-agent reinforcement learning is a sequence modeling problem") for a fairer evaluation.

_Learning curves of evaluation win rates are only provided for the proposed method in the ablation study. You need to include the evaluation return curves for both the proposed method and the baselines so the reader can actually understand the superior performance in terms of both higher return and sample efficiency.

Related work:

_The paper needs a much clearer distinction between model-free MARL, MBRL, and world models so the reader knows exactly where this method fits.

Minor comments:

_Throughout the text, \cite is used casually instead of \citep, so the names of the authors in the cited texts are randomly spread within the text and disrupt the flow. Please fix this.

_The first paragraph on page 4 explains many SMAC observation/action details that could be summarized more concisely with a citation to the SMAC paper. The manuscript does not need to describe hit points, cooldowns, terrain height, relative distances, and Boolean flags in so much detail unless those details are absolutely essential to your proposed tokenizer.

---

> ### Author Response · Authors · 2026-06-17
>
> We thank the reviewer for the positive assessment and for the concrete requests. First of all, we apologize for the delay in responding to the first draft. We were running several experiments to meet all the requested requirements.
>
> Following are the explanation of some modifications we made in our manuscript based on the requested changes:
>
> **1. Same codebase/version/protocol/budget.** We replaced the copied baseline table with a unified re-run of MACT, MARIE, MATWM, MAMBA, MBVD, MAPPO, and MAT. All methods use the same harness, pinned SC2.4.1.2.60604 build, and we have tried to follow all the standards proposed by the original methods. The revised table reports mean±std, and the aggregate analysis is now computed over all 12 evaluated maps. Following the requirements from another reviewer, we also increased the number of seed to 5 in all experiments.
>
> **2. Add a transformer-based sequence-modeling baseline.** We add Multi-Agent Transformer (MAT; Wen et al., 2022) as a model-free transformer policy baseline. Under the low-data protocol, MAT reaches an all-12-map aggregate IQM of 10.0, below the world-model methods. This clarifies that the observed sample efficiency is not just due to using a Transformer, but to learning and training inside a world model.
>
> **3. Add baseline learning curves.** We add per-map evaluation-win-rate curves with all methods overlaid (mean±std over seeds), plus an overview grid. Now it makes convergence speed and final performance directly comparable.
>
> **4. Clarify related work.** The Related Work section now starts with a taxonomy: model-free MARL (value factorization, policy gradient, and sequence-model policies such as MAT), model-based RL with online planning, and world-model methods that train policies by imagination without online joint-action search. We then place MACT precisely as a world-model method whose contribution is an AC-CPC objective.
>
> **5. Fix citation style.** We swept the manuscript to use parenthetical citations and reserve in-text citations for cases where author names are grammatically part of the sentence.
>
> **6. Condense environment details.** The enumeration of hit points, cooldowns, terrain height, distances, and flags has been replaced by a concise SMAC/Dec-POMDP summary with a citation to the SMAC paper. We kept only the information needed for the tokenizer.
>
> We have just uploaded a new version of the revised manuscript. Thank you again for your review.

---

> > ### Comment · Reviewer_pA2w · 2026-06-29
> > **Chosen hyper-parameters for MACT and the baselines**
> >
> > I appreciate addition of the new baseline and also the new learning curves. I still have a remaining concern.
> > Authors said: "we have tried to follow all the standards proposed by the original methods."
> > Were you able to replicate the results from those papers?
> > If not you should tune the hyper-parameters for each algorithm separately, and in a fair and consistent way for all algorithms.

---

> > > ### Author Response · Authors · 2026-06-30
> > >
> > > We thank the reviewer for the follow-up. We agree that our previous wording, “we tried to follow all the standards proposed by the original methods,” was too vague. To clarify: we did not use one shared hyperparameter setting across methods. Each baseline was configured separately, using the hyperparameters and map-specific settings from its own paper/repository whenever available. The shared part is the evaluation protocol: same SC2 build, same evaluation procedure, same revised seed count, and matched data budget within the low-data comparison.
> > >
> > > We also performed a published-budget sanity check for the two closest world-model baselines, MAMBA and MARIE. When run at their published budgets, MAMBA broadly reproduces its reported results, matching or exceeding the published numbers on most checked maps. MARIE also reproduces its reported behavior on several maps, including close agreement on `3m` and `3s_vs_3z`, and remains in the same high-performing regime on `3s_vs_5z`. Some MARIE cells, especially `8m` and `so_many_baneling`, remain below the published values, which we interpret as seed/environment sensitivity rather than a single under-tuned configuration. We report the **detailed recovery numbers** in our response to Reviewer 4wT9 to avoid duplicating the same table here.
> > >
> > > The key distinction is that our main table is not intended to be a direct 100K-step replication of the original baseline papers. It is a stricter low-data sample-efficiency comparison. For standard SMAC maps, our main comparison uses 50K environment steps, while several original papers report headline numbers at 100K steps. This lower-budget setting is deliberate because MACT is designed to improve sample efficiency under limited real-environment interaction. Therefore, lower absolute baseline values in the 50K table should not be read as evidence that the baselines were under-tuned. They are correctly configured baselines evaluated in a harder low-data regime.
> > >
> > > On *Appendix B: MACT and baselines implementation* (Table 4, 5, and 6), we explained in more details the hyperparameters we have used for each baseline re-training. We hope this clarifies that the baselines were tuned individually and consistently, and that MACT’s advantage in the main comparison reflects the stricter 50K sample-efficiency setting.

---

### Review · Reviewer_4wT9 · 2026-05-30

**Summary Of Contributions:**

The paper presents MACT, a decentralized Transformer world model for cooperative MARL. It builds directly on MARIE's CTDE design (a shared per-agent block-sparse Transformer over discretized vector-quantized observation and action tokens, with a single Perceiver cross-attention step producing per-agent global context) and augments MARIE's one-step token-reconstruction objective with an action-conditioned multi-step contrastive objective (AC-CPC), adapted from the single-agent method TWISTER.
The key design decisions the paper resolves for the multi-agent/CTDE setting are (i) what to predict — future team-conditioned Perceiver latents of an augmented future view, and (ii) what to condition on — each agent's own short action window rather than a team-aggregated joint-action summary.
On SMAC under a low-data budget, MACT attains the highest mean and median win rate over the reported baselines and shows its gains on coordination-heavy maps. The authors also conduct ablations covering the CPC horizon, observation-dropout strength, and per-agent vs. team-aggregated conditioning.

## Strengths

1. The motivation (one-step reconstruction biases representations toward short-term structure and drifts over long rollouts) is clear.
2. The per-agent vs. team-aggregated ablation is informative.
3. The architecture inherits MARIE's near-linear scaling in team size

## Weaknesses
1. First, the most critical issue is that the genuinely new component (AC-CPC) is a thin layer on top of MARIE. Most of *Section 3 Methodology* excluding the AC-CPC subsection re-describes MARIE's tokenizer, local Transformer, Perceiver aggregation, available action prediction, loss function, imagination loop, and behavior learning details in imagination, in places without clear attribution.
2. The empirical case for the central claim is incomplete: there is no controlled one-step (K_cpc=1) comparison against the MARIE baseline in the same codebase, no sample-efficiency curve comparison against baselines (curves are MACT-only, the results of baselines in Table 2 are copied from their papers), and no rollout-fidelity analysis.
3. Evaluation is confined to a subset of SMAC maps with only easy level of difficulty, and on the harder coordination maps MACT does not clearly improve over MARIE and MATWM. These are detailed under Requested Changes.

**Audience:**

Yes

**Audience Explanation:**

Sample-efficient multi-agent world models are an active topic in TMLR's audience.

**Broader Impact Concerns:**

No specific ethical concerns. The work is methodological and evaluated on a standard cooperative benchmark (SMAC), with no sensitive data or directly harmful application. A Broader Impact Statement is not required.

**Claims And Evidence:**

No

**Claims Explanation:**

The mechanism claims (intermediate CPC horizon is best; light observation dropout helps; per-agent beats aggregated conditioning) are supported by clean single-factor ablations and are convincing. However, two problems remains.

(1) The central claim — that action-conditioned multi-step prediction is what drives the gains over a one-step world model — is not isolated: the horizon sweep is $K \in \{ 4,8,12 \}$ and never includes the $K=1$, i.e., one-step point, so the gain cannot be attributed to AC-CPC rather than to implementation differences from MARIE.

(2) The main table copies baseline results from their original papers (different codebases/protocols/seed counts) and provides learning curves only for MACT, so the sample-efficiency comparison can be invalid and even tricky by taking the endpoint results while other baselines may exhibits faster convergence rate and more optimal convergence under sufficient environment steps.

**Requested Changes:**

**1. Restructure Section 3 to clearly separate what is adopted from MARIE from what is new — this is essential for an accurate contribution claim.** As written, the entire first part of the methodology (3.1 vector-to-token VQ-VAE; 3.2 local block-sparse Transformer; the Perceiver centralized aggregation, Eqs. 2–3; 3.3 one-step prediction heads and L_dyn; and 3.7 imagination-based actor–critic) is essentially a restatement of MARIE's architecture and training pipeline. Some paragraphs do say "Following MARIE," but several of the largest ones (tokenizer, local Transformer, Perceiver aggregation) do not attribute clearly, so a reader can easily mistake adopted components for contributions of this paper.

**2. Add the one-step (K_cpc=1) controlled ablation against MARIE.** Run MACT with K_cpc=1 and MARIE in the same implementation, protocol, and map, to validate the paper's central claim.

**3. Re-running the baseline would be better to demonstrate the validity of current results.** Baseline numbers in Table 2 are taken from the respective papers. It would be better to re-run at least MARIE and MATWM under your pinned SC2.4.1.2.60604 build and codebase and report your numbers alongside the published ones. Besides, it would be better to state clearly imagined-rollout budgets across MACT and other baselines.

**4. Provide a controlled sample-efficiency comparison curve.** Currently learning curves are given only for MACT. Once MARIE and MATWM are re-run under identical experimental conditions (see point 3), overlaying its learning curve against MACT on representative maps would make the sample-efficiency claim directly verifiable in a single plot.

**5. Provide rigorous statistical testing.** This is critical given the small and inconsistent number of seeds (3–4), which raises concerns about the reliability and significance of the reported gains. Adopting rigorous RL evaluation protocols such as those outlined in rliable_code and rliable_paper would strengthen the empirical claims.

**6. Compare MACT with other baselines on more extensive SMAC maps and benchmarks.** Current evaluation is only limited to a SMAC subset with easy level of difficulty (only 3s_vs_5z map is of hard level). Evaluation on harder maps (e.g., corridor, 6h_vs_8z) and a second benchmark (e.g., GRF, MA-MuJoCo) would better demonstrate the general improvement of MACT—especially since the closest prior work, MARIE, reports results on 13 SMAC maps (including SuperHard corridor) as well as MA-MuJoCo and SMACv2.

**7. Add an empirical compute comparison.** Appendix B gives a theoretical complexity analysis. Adding wall-clock time and memory usage comparison for MARIE vs. MACT would better support the analysis.

**8. Add a rollout-fidelity / multi-step prediction-accuracy analysis.** The paper argues AC-CPC yields more consistent long-horizon rollouts and mitigates non-stationarity/credit assignment. A compounding-error-vs-horizon comparison of MACT vs. MARIE would substantiate this.

---

> ### Author Response · Authors · 2026-06-17
>
> We thank the reviewer for the detailed critique. We agree that the original paper did not isolate the AC-CPC contribution clearly enough and did not make the sample-efficiency comparison verifiable. The revision addresses these issues directly.
>
> **Foundational change.** We re-ran MACT and all baselines (MARIE, MATWM, MAMBA, MBVD, MAPPO, and the newly added MAT baseline) in one harness with a pinned SC2.4.1.2.60604 build, and identical per-map environment-step budgets. This replaces the previous copied baseline table.
>
> **1. Separate adopted MARIE components from the new contribution.** Section 3 now explicitly states that MACT uses the MARIE tokenized per-agent Transformer/Perceiver backbone: the vector tokenizer, local Transformer dynamics, Perceiver aggregation, one-step prediction heads, and imagination-based actor-critic are adopted from that pipeline and restated only for self-contained notation. The manuscript then identifies MACT's contribution as the AC-CPC objective added on top of this backbone, including the two CTDE-specific choices: predicting future team-conditioned Perceiver latents and conditioning on each agent's own action window. We also updated the method-comparison table caption and the introductory framing so readers do not mistake the adopted backbone for a claimed architectural contribution.
>
> **2. Add the K_CPC=1 control.** We add the one-step control to the horizon ablation. On 3m, K=1 learns early but later collapses, whereas multi-step horizons remain stable. This shows that the gain is not from adding an arbitrary contrastive head, but from multi-step action-conditioned prediction.
>
> **3. Re-run MARIE/MATWM and state rollout budgets.** MARIE and MATWM are re-run in the same harness as all other baselines. The unified table reports mean±std under the same evaluation protocol. We also add appendix tables for imagination horizon, imagined-trajectory batch, collection interval, world-model batch, update counts, total budget, and per-map overrides.
>
> **4. Add controlled sample-efficiency curves.** We add overlaid evaluation-win-rate curves for all methods on every map, plus an overview grid. Since the methods now share the same per-map budget, the curves show both convergence and endpoint performance under one protocol.
>
> **5. Add rigorous statistical testing.** We add rliable-style aggregate metrics over all 12 evaluated maps. The IQM scores are MACT **80.4**, MAMBA **60.9**, MARIE **52.1**, MATWM **14.1**, MAPPO **11.3**, MAT **10.0**, and MBVD **3.8**. The revised appendix figure reports the 95% bootstrap intervals, performance profiles, and P(MACT > X).
>
> **6. Harder maps and second benchmark.** We added two SuperHard SMAC maps, corridor and 6h_vs_8z, and used the revision period to expand the same-protocol comparison toward a standardized five-seed evaluation. Given the available revision time, we prioritized making the SMAC evaluation broader and more reliable rather than launching a full second benchmark suite. We therefore did not have enough time to run GRF or MA-MuJoCo, and we now state this explicitly as a limitation and future-work direction.
>
> **7. Compute comparison.** We add a MARIE-vs-MACT world-model-update benchmark. AC-CPC adds about 0.6GB peak memory and update-time overhead that decreases with map size (+75% on 3m, +26% on MMM), so we soften the scalability claim accordingly.
>
> **8. Rollout fidelity.** We add teacher-forced rollout-fidelity analysis on so_many_baneling, MMM, and 3s_vs_3z. MACT's observation error grows more slowly than MARIE's, especially on 3s_vs_3z (below 0.8 vs. about 4.5 at horizon 15), while reward error is comparable. We therefore claim improved multi-step state/observation consistency, not improved reward prediction.
>
> Thank you again for your review, and please feel free to see the updates already included in the new manuscript version.

---

> ### Comment · Reviewer_4wT9 · 2026-06-25
>
> I thank the authors for a thorough and responsive revision. While my concerns were mostly addressed, one issue remains.
>
> **Baseline comparison remains in question**. Re-running all baselines under one harness was the right step and I appreciate it. However, the re-run appears to under-tune the transformer baselines relative to their published configurations, which would make the comparison unfair rather than genuine weakness in this regime. Concretely, Table 3 in Appendix of current revised manuscript lists the world-model Transformer at 2 layers for both MACT and MARIE, whereas MARIE's own paper uses a 10-layer shared Transformer (Appendix A.2, Table 3). Reducing a 10-layer backbone to 2 layers would substantially curtail exactly the long-horizon modeling capacity MARIE relies on.
> A related symptom appears with MATWM, whose performance drops to 0.0 on 8m, so_many_baneling, and 3s_vs_5z at the same 50K/200K budgets and game backend SC2.4.1.2.60604 used in its own paper (67.0 / 86.0 / 64.0), which matches MATWM's own ablation where these maps collapse to 0.0 when its teammate predictor is inactive.
> It would be better to verify a few baselines (e.g., MAMBA, MARIE) against their published same-budget numbers and report whether the harness recovers them, and describe the tuning effort applied per baseline.

---

> > ### Author Response · Authors · 2026-06-29
> >
> > We thank the reviewer for the careful follow-up. We agree that, if MARIE or MACT had actually been run with a 2-layer world-model Transformer, the comparison would be unfair. We therefore audited the launch configurations, code paths, and saved experiment metadata. The baselines were not run with a 2-layer world-model Transformer. The ''2'' in Appendix Table 3 was a reporting error. It referred to the depth of the Perceiver aggregation module, not to the local world-model Transformer.
> >
> > The corrected architecture specification is:
> >
> > | Component | Layers / depth | Heads | Embed dim |
> > |---|---:|---:|---:|
> > | World-model Transformer / local dynamics | **10** | 4 | 256 |
> > | Perceiver aggregation module | **2** | 8 | 256 |
> >
> > Thus, MARIE was not capacity-reduced in our harness. We apologize for the confusing table entry and will correct Appendix Table 3 accordingly.
> >
> > We also performed a baseline verification. We re-ran MAMBA and MARIE against their published same-budget settings. For MARIE, we used the published map-dependent configuration and the published map-specific overrides, including temperature = 2.0 and collection interval = 200 on `3s_vs_5z`. For MAMBA, we used the upstream published configuration, with only evaluation/logging and cluster-execution plumbing changed.
> >
> > The recovery check is:
> >
> > | Map / budget | MAMBA — ours / published | MARIE — ours / published |
> > |---|---:|---:|
> > | `3m` / 100K | 94.5 / 87.7 | 98.9 / 99.5 |
> > | `8m` / 100K | 91.1 / 50.2 | 69.4 / 88.0 |
> > | `3s_vs_3z` / 100K | 97.6 / 89.3 | 98.0 / 98.9 |
> > | `so_many_baneling` / 100K | 92.0 / 91.6 | 79.8 / 94.8 |
> > | `3s_vs_5z` / 200K | 15.0 / 13.4 | **66.5±16.4 / 78.4±11.2** |
> >
> > These results show that the harness broadly recovers the published behavior of MAMBA and MARIE. MAMBA matches or exceeds its published result on most checked maps, including `3s_vs_5z`. MARIE matches closely on `3m` and `3s_vs_3z`, and on `3s_vs_5z` its all-seed result remains in the same high-performing regime as the published result with exception of `8m` and `so_many_baneling`.
> >
> > For `3s_vs_5z`, we observed strong seed sensitivity. One MARIE seed obtained only **0.9%** final win rate, including such a collapsed seed substantially lowers the mean on this environment. Our revised evaluation uses 5 seeds, so a single collapsed seed can visibly affect the average. We will explicitly describe this map as high-variance rather than treating it as a clean MARIE failure.
> >
> > About MATWM, we agree that the 0.0 cells require careful interpretation. We do not intend to claim that MATWM is generally weak. We attempted to contact the MATWM authors through their GitHub repository to clarify some reproducibility issues, but the project's repository later became unavailable, so we could not obtain further confirmation. However, we can guarantee that we followed all the training guidelines from both paper and readme instructions.
> >
> > We also clarify that the main comparison in the revised manuscript is intentionally a **lower-data sample-efficiency comparison**. For the standard SMAC maps, our main table uses **50K environment steps**, whereas several original baseline papers report their headline numbers at **100K steps**. Same thing for `corridor`, where the standard training steps budget are 400K and we used 200K. This lower-budget setting is deliberate, MACT is designed to improve sample efficiency under limited real environment interaction. Therefore, lower absolute baseline values in our 50K table should not be interpreted as evidence that the baselines were under-tuned. They are just being evaluated in a stricter low-data regime.

---

> > > ### Comment · Reviewer_4wT9 · 2026-06-30
> > >
> > > Thank you for the clarifications and the recovery check. My main concerns have been addressed.

---

### Review · Reviewer_hBP8 · 2026-06-04

**Summary Of Contributions:**

The paper proposes MACT, a decentralized transformer-based world model for cooperative MARL under CTDE. Its main contribution is an action-conditioned multi-step contrastive objective, where each agent predicts future team-conditioned latent representations from its local history and short future action window. The learned world model is further used for latent imagination with decentralized actors and a centralized critic. Empirically, MACT achieves the best reported performance on SMAC, with clear gains over MATWM, MARIE, MAMBA, MBVD, and MAPPO, especially on coordination-heavy maps. The ablations support the usefulness of the CPC horizon, light observation dropout, and per-agent action conditioning.

The key strength is that MACT combines scalable token-based multi-agent world modeling with action-conditioned predictive representation learning, while avoiding explicit online planning over the joint action space. A main weakness is that the evaluation is limited to SMAC, so the generality of the method across more diverse multi-agent domains remains unclear.

**Additional Comments:**

The paper is promising and the main idea is coherent. I especially like that the authors do not claim universal superiority: the discussion of MATWM's advantage on micro-intensive maps is honest and useful. My main concern is that the empirical evidence needs to be tightened before the paper can fully support its conclusions. A stronger same-protocol comparison and more transparent uncertainty reporting would make the contribution much easier to trust.

**Audience:**

Yes

**Audience Explanation:**

MACT targets an important problem: improving sample efficiency in MARL without relying on expensive online planning over the joint action space. The paper is also timely, as it connects transformer-based world modeling, contrastive predictive learning, and CTDE-style MARL. Although the evaluation could be broader and the ablations more comprehensive, the reported gains on coordination-heavy SMAC maps and the weaker results on some micro-intensive maps are both informative.

**Broader Impact Concerns:**

I do not see major immediate broader-impact concerns in the current submission, since the experiments are on SMAC and the contribution is methodological.

**Claims And Evidence:**

Yes

**Claims Explanation:**

The main claim is mostly supported by the reported SMAC results, especially in the low-data setting, and the paper clearly states that its evidence is limited to structured-observation SMAC with discrete actions.

However, the evidence is not fully convincing. Some baseline results are taken from prior papers rather than rerun under a unified pipeline, and MACT is close to or worse than MATWM on several maps. The variance is also large on some tasks, such as MMM. In addition, the ablations are limited to 3m and 8m, so they do not fully explain the behavior on harder or underperforming maps.

**Requested Changes:**

1. The baseline comparison should be strengthened. The current table relies on numbers from separate prior papers. Please either rerun the strongest baselines, at least MARIE and MATWM, under the same codebase and evaluation protocol, or provide a much more detailed justification that the reported numbers are directly comparable.

2. The paper should report uncertainty more transparently. Please include per-seed results or confidence intervals, especially for high-variance maps such as MMM. The paper should avoid overstating wins when the variance overlaps with competitive baselines.

3. The ablations should be broadened, or the claims should be narrowed. The CPC horizon study is on 3m, while the paper uses K_cpc=5 on so_many_baneling and 2s3z; the dropout and action-conditioning ablations are on 8m. Please explain why these maps are sufficient, or add ablations on at least one map where MACT trails MATWM, such as 2s3z or 3s_vs_4z. This would clarify whether the design choices fail on micro-intensive scenarios or whether the current hyperparameters are simply not tuned for them.

4. The practical scalability claim should be supported with measurements. The paper states near-linear complexity in the number of agents, but the experiments are all SMAC maps with modest team sizes. Please add wall-clock, memory, or synthetic scaling measurements as the number of agents or tokens increases, or soften the practical scalability claim.

5. The analysis of failure cases should be improved. MACT underperforms MATWM on 2s3z, 3s_vs_3z, and 3s_vs_4z. The hypothesis that MATWM's teammate predictor gives better short-horizon cross-agent cues is plausible, but currently speculative. Please add qualitative rollout analysis, attention/context diagnostics, or an ablation with additional Perceiver passes or richer cross-agent context to make this discussion more convincing.

---

> ### Author Response · Authors · 2026-06-17
>
> We thank the reviewer for the careful and constructive assessment. We agree that the main issue was empirical trust: the original submission mixed cross-paper baseline numbers with MACT curves, reported limited uncertainty, and discussed failure cases partly speculatively. The revision was organized around fixing that.
>
> **Unified evaluation.** We re-ran MACT and six baselines (MARIE, MATWM, MAMBA, MBVD, MAPPO, and the newly added MAT baseline) in one harness with a pinned SC2.4.1.2.60604 build, and trying to follow as much as possible the original hyperparameters used in each paper. The revised table reports mean±std, and the aggregate analysis now uses all 12 evaluated maps (as requested by another reviewer we increased two very hard environments to the benchmark).
>
> **1. Stronger baseline comparison.** Table 2 is replaced by the unified-protocol re-run (now by a fixed number of 5 different seeds). This materially changes the comparison: under the pinned build, MATWM does not reproduce the copied published numbers and is no longer the closest competitor. On the all-12-map aggregate IQM, MACT scores **80.4**, ahead of MAMBA **60.9**, MARIE **52.1**, MATWM **14.1**, MAPPO **11.3**, MAT **10.0**, and MBVD **3.8**. We include the stratified-bootstrap intervals in the revised appendix figure rather than in the abstract-level summary.
>
> **2. Uncertainty and high-variance maps.** Every table cell now reports mean±std. We add rliable-style IQM/median/mean CIs over all 12 maps, performance profiles, and probability of improvement. We also soften the map-level language: on MMM, MAMBA is best in the unified table and MACT is not claimed as a clear per-map win, the main claim rests on the aggregate IQM.
>
> **3. Broader and narrower ablations.** We add the requested one-step control: K_CPC=1 initially learns on 3m but later collapses, whereas multi-step horizons remain stable. We also move the augmentation ablation to the harder MMM map. Importantly, the updated result shows augmentation is map-sensitive: no augmentation is strongest on MMM, so we now phrase augmentation as a tunable regularizer rather than a universal improvement. The per-agent conditioning ablation remains strongly positive on 8m.
>
> **4. Scalability and compute.** We add a controlled MARIE-vs-MACT world-model-update benchmark. AC-CPC adds about 0.6GB peak memory (roughly 8%) and a relative update-time overhead that shrinks with map size (+75% on 3m, +26% on MMM). We therefore soften the claim to "near-linear backbone with modest AC-CPC overhead."
>
> **5. Failure cases and diagnostics.** We remove the speculative claim that MATWM's teammate predictor explains MACT's micro-map failures, because MATWM no longer dominates under the fair re-run. Instead, we add teacher-forced rollout-fidelity diagnostics against MARIE. On 3s_vs_3z at horizon 15, MACT's observation error remains below 0.8 while MARIE reaches about 4.5. Reward error is comparable, so we phrase the conclusion as improved multi-step state/observation consistency rather than lower reward error.
>
> Please feel free to see the modification in the new updated manuscript. Thank you again for your insightful review.